

# Retrospective study of a novel hematological parameter for predicting the survival of patients with nasopharyngeal carcinoma

Wenhua Tang[1] and Guoxian Long[2]

[1] Department of Oncology and Southwest Cancer Center, Southwest Hospital, Army Medical University, Chongqing, Chongqing, China
[2] Department of Oncology, Tongji Hospital, Tongji Medical College, Huazhong University of Science and Technology, Wuhan, Hubei, China

Corresponding authors
Wenhua Tang,
tangwenhua@tmmu.edu.cn
Guoxian Long, oncolong@163.com,
374994698@qq.com

## ABSTRACT

**Purpose**. This study aims to explore the prognostic values of routine pre-treatment hematological parameters in patients with nasopharyngeal carcinoma (NPC).

**Methods**. The hematological parameters and clinical data of patients with NPC were collected from January 2012 to December 2013 at Tongji Hospital, Tongji Medical College, Huazhong University of Science and Technology. The survival statistics were obtained by regularly following-up the patients. The cut-off values for the hematological parameters were calculated using X-tile software. SPSS version 24.0 was used for the statistical analysis. The relationship between the hematological parameters and the prognosis of patients with NPC was analyzed using the Kaplan–Meier method and Cox multivariate regression. The discriminating abilities of the factors, which predict the prognosis, were evaluated by utilizing the receiver operating characteristic (ROC) area under the curve (AUC).

**Results**. This study included 179 patients with NPC. Multivariate analysis shows that pretreatment platelet-to-lymphocyte ratio (PLR; hazard ratio; HR = 0.44, 95% CI [0.21–0.91], $p = 0.029$), serum albumin (ALB; HR = 2.49, 95% CI [1.17–5.30], $p = 0.018$), and globulin (GLO; HR = 0.44, 95% CI [0.21–0.90], $p = 0.024$) are independent predictors for 5-year overall survival (OS) in patients with NPC. In addition, pre-treatment PLR (HR = 0.47, 95% CI [0.25–0.90], $p = 0.022$) and pre-treatment GLO (HR = 0.37, 95% CI [0.19–0.72], $p = 0.001$) are associated with 5-year progression-free survival (PFS) in patients with NPC. Based on the results of the multivariate analysis, we proposed a new biomarker GLO-PLR, which is observably correlated with the T stage, N stage and clinical stage in patients with NPC. The OS resolving ability of the GLO-PLR evaluated by AUC is 0.714, which is better than those of GLO and PLR. The PFS resolving ability of the GLO-PLR evaluated by AUC was 0.696, which is also better than those of GLO and PLR.

**Conclusion**. Pre-treatment PLR, ALB, and GLO are independent predictors of 5-year OS in patients with NPC, where PLR and GLO are also independent predictors of 5-year FPS. Compared with other hematological parameters, the proposed GLO-PLR is an inexpensive, effective, objective, and easy-to-measure marker for predicting the prognosis of NPC.

## INTRODUCTION

Nasopharyngeal carcinoma (NPC), a common cancer of the head and neck, has a high prevalence in non-Western countries: approximately 70% of new cases occur in China, East Asia, Southeast Asia, the Middle East, and North Africa (*Chen et al., 2019*; *Bray et al., 2018*). Due to its sensitivity, radiotherapy is the primary treatment modality for patients with NPC. In particular, intensity-modulated radiotherapy (IMRT) has been widely used for NPC treatment. Compared with conventional three-dimensional conformal radiotherapy, IMRT has better conformity and dose distribution in the target (*Moon et al., 2016*). It can deliver a lower dose to the normal tissues that surround the tumor and a higher dose to the tumor (*Nutting et al., 2011*). Therefore, it can significantly improve the survival and local control rates while reducing the adverse reactions. Guidelines recommend that patients with stage I disease receive radiotherapy alone, and those with stage II-IVB disease receive concurrent chemoradiotherapy and/or adjuvant or neoadjuvant chemotherapy (*National Comprehensive Cancer Network (NCCC), 2024*). There are some side effects during treatment, such as hematological side effects, nausea, vomiting, xerostomia, subcutaneous fibrosis, hearing impairment, temporal lobe necrosis, and weight loss (*Li et al., 2022*). With standard treatment, the 5-year overall survival (OS) rate in patients with NPC can reach 75% (*Lee et al., 2015*). However, 20–30% of patients still have the risk of distant metastases and/or local recurrences with poor prognosis (*Chen et al., 2012*; *Sun et al., 2014*). Currently, treatment strategies and prognosis for patients with NPC mostly depend on their staging according to the tumor-node-metastasis (TNM) system (*Pan et al., 2016*). However, patients with NPC in the same TNM stage often have a different clinical course (*Au et al., 2003*). The possible explanation is that the TNM staging system classifies the disease extent without considering the biological heterogeneity, and it is primarily based on anatomical information of the tumor. Therefore, it is of great significance to discover molecular predictors that can supplement TNM staging to predict the prognosis. Molecular biomarkers can also help predict the recurrence and metastasis risk in patients with NPC and guide the selection of better therapeutic strategies that could improve survival rates.

Several molecular biomarkers have been considered as prognostic factors for NPC, including, EBV mRNA-interfering complementary RNAs (micRNAs), Epstein-Barr virus (EBV) DNA load, vascular endothelial growth factor (VEGF), cyclooxygenase-2 (COX-2), and epidermal growth factor receptor (EGFR) overexpression (*Pan et al., 2013*; *Li et al., 2023*; *Zhang et al., 2015*). However, before these biomarkers are included in routine testing, their elevated costs and high variability of results among different laboratories should be considered. Therefore, there is an urgent need to explore some inexpensive, simple, and objective predictors to predict the prognosis of patients with NPC.

Inflammatory mediators are important constituents of the tumor microenvironment (*Mantovani et al., 2008*). Some researchers believe that systemic inflammatory reaction play a critical role in tumor occurrence and development (*Hanahan & Weinberg, 2011*). At present, some hematological parameters of the systemic inflammatory reaction have become independent predictors for patients with cancer. These include white blood

cell count (*Pei et al., 2014*), monocyte count (*Tsai et al., 2014*), lymphocyte count (*Feng et al., 2018*), platelet count (*Ishizuka et al., 2013*), neutrophil-to-lymphocyte ratio (NLR) (*Yao et al., 2019*), platelet-to-lymphocyte ratio (PLR) (*Feng, Huang & Chen, 2014*), lymphocyte-to-monocyte ratio (LMR) (*Lin et al., 2014*), Glasgow prognosis score (GPS) (*Gao & Huang, 2014*), and neutrophil-to-monocyte ratio (NMR) (*Tsai et al., 2014*). The measurements of these biomarkers are easy, non-invasive, fast, inexpensive, and highly reproducible. Previous studies have reported that PLR and LMR can predict the prognosis of NPC patients who receive conventional radiotherapy (*Sun et al., 2016*; *Lu et al., 2017*). It has also been found that these two predictors can predict the prognosis of NPC patients who receive IMRT (*Jiang et al., 2018*). However, most of these studies only evaluated one predictor, which makes the statistics of their results less potent.

Albumin (ALB) and globulin (GLO) are the two main constituents of serum proteins and function as representative indicators of systemic inflammation (*Fernández et al., 2019*; *Gabay & Kushner, 1999*). ALB levels reflect the nutritional status of patients (*Kirsch et al., 1968*). Previous studies have demonstrated that serum ALB has prognostic significance in several kinds of cancers, including lung cancer, gastrointestinal cancer, breast cancer, ovarian cancer, and NPC (*Gupta & Lis, 2010a*; *Li et al., 2014*). In a prospective study containing 4231 patients, the biochemical signature, which is a combination of six hematological parameters including albumin, can predict the prognosis of patients with NPC (*Sun et al., 2022*). GLO also plays an important role in inflammation and immunity and acts as a carrier of sex hormones (*Gabay & Kushner, 1999*). No studies have assessed the predictive value of GLO for OS in patients with NPC, and only one study has assessed the predictive value of ALB/GLO for OS in patients with NPC. However, that study was based on survival data from patients who receive conventional radiotherapy. Our study attempted to assess the influence of hematological parameters as clinical factors on the prognosis of patients who received IMRT. We proposed a new hematological parameter termed as GLO-PLR, which is a combination of two independent prognostic predictors. It is an inexpensive, effective, objective, and easy-to-measure marker for predicting the prognosis of NPC patients.

## MATERIAL AND METHODS

### Study design

In this retrospective study, we attempted to assess the influence of PLR, LMR, hemoglobin (HB), GLO, and ALB as clinical factors on the prognosis of patients who received IMRT. We enrolled 179 patients who were treated at the Department of Oncology, Tongji Hospital, Tongji Medical College, Huazhong University of Science and Technology, between January 2012 and December 2013. For the first 3 years, all patients were required to return to the hospital for examination every 3 months. Then, the examination is required every 6 months for the next two years. After the completion of treatment, the examination are required annually. The time from the day of treatment to the day of death or November 2020 is regarded as the duration of follow-up. We received a waiver of informed consent during follow-up. The study was approved by the Medical Ethics Committee of the Tongji Hospital of Huazhong University of Science and Technology (TJ-IRB20211276).

## Inclusion criteria

The inclusion criteria are as follows: (1) histologically confirmed as NPC; (2) no proof of distant metastasis; (3) no previous malignancies; (4) no previous therapy for cancer; (5) full record of pre-treatment laboratory blood test results, including ALB and GLO levels; (6) no other serious diseases that could affect the results of the blood tests (acute or chronic infection, blood disease, autoimmune disease, pulmonary infarction, congestive heart failure, acute or chronic renal disease); (7) no drugs that could affect the results of blood tests, used at least 1 month before treatment; (8) venous blood taken during fasting or at least 5 h after the last meal, from resting participants; (9) complete follow-up data; (10) successful radiotherapy with or without chemotherapy; (11) age is older than 18; and (12) gender is not limited. We obtained the patient's clinical data from the medical records. Finally, 179 patients with NPC who fulfilled the basic criteria are included. The age was divided by 50 years old. The patients in stage I, II, III, IVa and IVb were counted separately. In this study, routine workups accomplished before the start of therapy involving hematological parameters testing, endoscopy of the nasopharynx, magnetic resonance imaging or computed tomography scan of the neck and nasopharynx, whole-body bone imaging, abdominal ultrasonography, chest radiography, and physical examination. The seventh edition of the Union for International Cancer Control/American Joint Committee on Cancer (UICC/AJCC) staging system was adopted (*Sobin, Gospodarowicz & Wittekind, 2009*; *Compton et al., 2010*). According to our treatment guidelines, patients with stage I disease receive radiotherapy alone, and those with stage II-IVB disease receive concurrent chemoradiotherapy and/or adjuvant chemotherapy or neoadjuvant chemotherapy. The nasopharyngeal region receive a total planned dose of 68–70 Gy and involved cervical node receive the dose of 60–66 Gy. The regimens for neoadjuvant chemotherapy included TPF (docetaxel 75 mg/m2 intravenously (IV) on day 1, cisplatin 25 mg/m2 IV on days 1–3, 5-fluorouracil 750 mg/m2 continuously IV on days 1–5) and TP (docetaxel 75 mg/m2 IV on day 1, cisplatin 25 mg/m2 IV on days 1–3). These treatments were repeated every 21 days for 2–3 cycles. During radiotherapy, concurrent chemotherapy was performed at the same time (cisplatin 40 mg/m2 IV every week or 100 mg/m2 IV every 21 days). For patients who accepted adjuvant chemotherapy, the PF regimen (cisplatin 25 mg/m2 IV on days 1–3, 5-fluorouracil 750 mg/m2/d continuously IV on days 1–5) or TP regimen (docetaxel 75 mg/m2 IV on day 1, cisplatin 25 mg/m2 IV on days 1–3) were repeated every 21 days for 2–4 cycles.

## Parameters evaluated

PLR was the division of platelet counting by the lymphocyte count, whereas LMR was the division of lymphocyte counting by the monocyte count. The first outcome of this study is OS, which is defined as the time from NPC diagnosis to death from any cause. The second outcome of this study is progression-free survival (PFS), which is defined as the time from the diagnosis of NPC to tumor progression for any aspect or death from any reason. Patients who were alive or without progression were censored on the day of last contact.

## Determination of cut-off values

The best cut-off values of hematological parameters for the prognoses of NPC were computed by the X-tile software, which is a bio-informatics tool for biomarker assessment and outcome-based cut-point optimization (*Camp, Dolled & Rimm, 2004*). The X-tile software obtains cut-points by defining partitions in the "training set". They were then validated in a separate cohort of patients (the "validation set"), enabling a rigorous statistical evaluation. The X-tile software akso provides a method of dividing a single cohort into training and validation subsets for *P* value estimation. In this study, the survival time corresponding to different hematological parameters are selected as the cohort, which will be used for the cut-off value calclulation in X-tile software.

## Statistical analysis

In this study, survival analysis were performed using Kaplan–Meier analysis with the log-rank test. The Cox proportional hazards model was used for univariate and multivariate analyses. All statistical tests were two-sided, and statistical significance was set at $p < 0.05$. The discriminatory abilities of the prognostic factors were evaluated using the receiver operating characteristic (ROC) area under the curve (AUC). The statistical analyses were executed using IBM SPSS software (version 24.0; IBM Corporation, Armonk, NY, USA).

# RESULTS

## Patient characteristics

Table 1 demonstrates the clinical characteristics of the 179 patients with NPC who fulfilled the basic criteria. A total of 136 (76.0%) male patients and 43 (24.0%) female patients were included, with a sex ratio of 3.08:1. The mean age of these patients is 46 years (20–74 years), and 52 (29.1%) patients were more than 50 years old when diagnosed. The stage at diagnosis was I in one patient (0.6%), II in 19 patients (10.6%), III in 68 patients (38.5%), and IVA/IVB in 90 (50.3%) patients. 169(94.4%) patients were treated with concurrent radiochemotherapy, and 10 (5.6%) patients received radiotherapy alone. The median follow-up duration is 50.9 months for OS and 51.8 months for PFS. Thirty-four patients died. Fourteen patients were lost to follow-up. The proportion of patients lost to follow-up is 7.8% (<10%).

## Determine the best critical values

The optimal cut-off values of PLR, LMR, HB, ALB, and GLO for the prognosis of patients with NPC are demonstrated in Fig. 1. The best cut-off value is 130.7 ($p = 0.035$) for PLR, 3.3 ($p = 0.013$) for LMR, 14.2 g/dL ($p = 0.186$) for HB, 37.2 g/L ($p = 0.002$) for ALB, and 35.8 g/L ($p = 0.003$) for GLO.

## Univariate and multivariate analyses

Univariate analyses were executed using age, sex, T stage, N stage, clinical stage, PLR, MLR, ALB, and GLO as possible variables (Table 2). N stage ($p = 0.003$), PLR ($p = 0.001$; Fig. 2A), LMR ($p = 0.001$; Fig. 2B), HB ($P = 0.010$; Fig. 2C), ALB ($P < 0.001$; Fig. 2D), and GLO ($P < 0.001$; Fig. 2E) are independent predictors for lower five-year OS rates in

Table 1 Clinical characteristics of the patient population ($n = 179$).

| Characteristics | Number of patients | Percentage (%) |
|---|---|---|
| **Gender** | | |
| Male | 136 | 76.O |
| Female | 43 | 24.0 |
| **Age at diagnosis (years)** | | |
| ≤50 | 127 | 70.9 |
| >50 | 52 | 29.1 |
| **Histological subtype** | | |
| Squamous cell carcinoma | 15 | 8.4 |
| Non-keratinizing carcinoma | 84 | 46.9 |
| Undifferentiated carcinoma | 80 | 44.7 |
| **Tumor stage** | | |
| T0 | 1 | 0.6 |
| T1 | 17 | 9.5 |
| T2 | 60 | 33.5 |
| T3 | 49 | 27.4 |
| T4 | 52 | 29.1 |
| **Node stage** | | |
| N0 | 20 | 11.2 |
| N1 | 39 | 21.8 |
| N2 | 75 | 41.9 |
| N3 | 45 | 25.1 |
| **UICC/AJCC stage** | | |
| I | 1 | 0.6 |
| II | 19 | 10.6 |
| III | 69 | 38.5 |
| IVA/IVB | 90 | 50.3 |
| **Therapy** | | |
| Radiotherapy | 10 | 5.6 |
| chemoradiotherapy | 169 | 94.4 |
| **Survival** | | |
| Death | 34 | 19.0 |
| survival | 131 | 73.2 |
| loss to follow-up | 14 | 7.8 |

**Notes.**
UICC/AJCC, Union for International Cancer Control/American.

patients with NPC. The multivariate analyses reveal that a PLR > 130.7 (hazard ratio (HR) = 0.44, 95% Confidence Interval (CI) [0.21–0.91], $p = 0.029$), ALB ≤37.2 g/L (HR = 2.49, 95% CI [1.17–5.30], $p = 0.018$), and GLO > 35.8 g/L (HR = 0.44, 95% CI [0.21–0.90], $p = 0.024$) are independent predictors of lower five-year OS rates in patients with NPC (Table 3). In addition, univariate and multivariate analyses were also used for PFS analysis. The significant prognostic predictors identified by univariate analyses include N stage ($p = 0.001$), clinical stage ($p = 0.045$), PLR ($p = 0.002$; Fig. 3A), LMR ($p = 0.003$; Fig. 3B), HB ($p = 0.067$; Fig. 3C), ALB ($p < 0.001$; Fig. 3D), and GLO ($p < 0.001$; Fig. 3E), which are
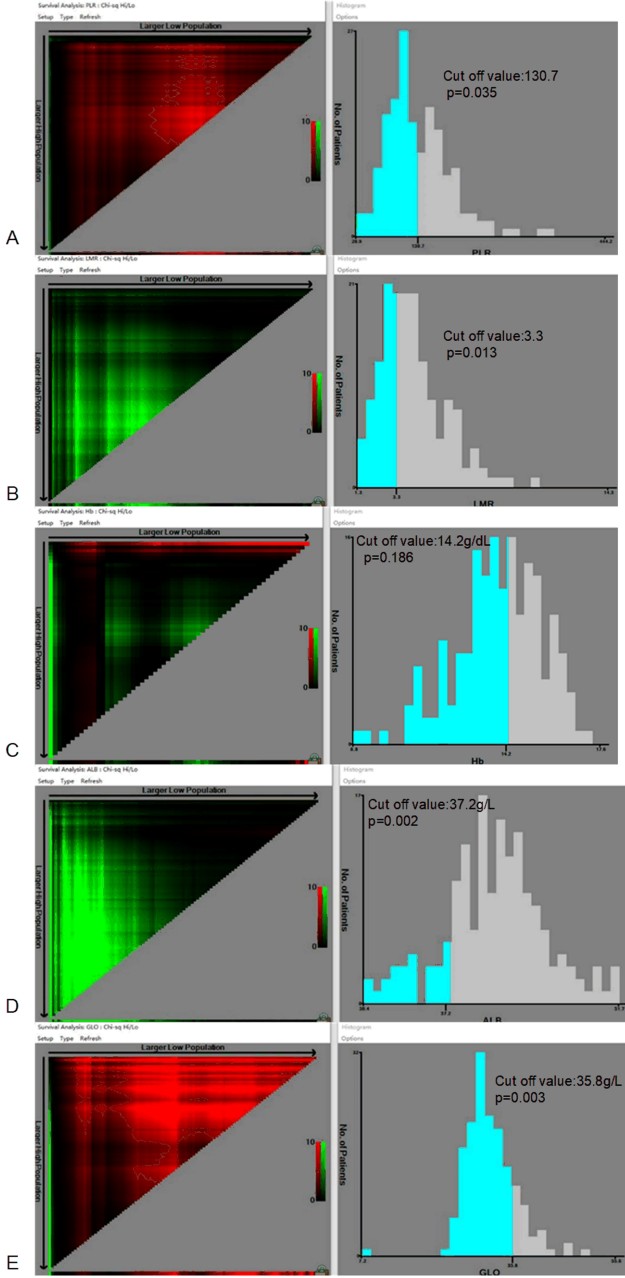

**Figure 1  Calculation of the cut-off values for the platelet-to-lymphocyte ratio (PLR), lymphocyte-to-monocyte ratio (LMR), hemoglobin (HB), albumin (ALB), and globulin (GLO) using the X-tile software.**

also independent predictors for lower five-year OS rates in patients with NPC (Table 4). Multivariate analyses show that PLR > 130.7 (HR = 0.47, 95% CI [0.25–0.90], $p = 0.022$), and GLO > 35.8 g/L (HR = 0.37, 95% CI [0.19–0.72], $p = 0.001$) are independent predictors of lower five-year PFS rates in patients with NPC (Table 5).

**Table 2** Analysis of clinicopathological characteristic for the prediction of overall survival in patients with nasopharyngeal carcinoma.

| Characteristics | No. of patients | S.T. (Month) Mean (95%, CI) | p-value |
|---|---|---|---|
| **Sex** | | | 0.291 |
| Male | 134 | 55.31 (51.39–59.24) | |
| Female | 45 | 53.08 (50.42–55.74) | |
| **Age at diagnosis, (years)** | | | 0.205 |
| ≤50 | 52 | 56.30 (53.19–59.42) | |
| >50 | 127 | 52.53 (49.68–55.38) | |
| **Tumor classification** | | | 0.297 |
| T0-T2 | 78 | 55.00 (52.04–57.95) | |
| T3-T4 | 101 | 52.59 (49.38–55.80) | |
| **Node classification** | | | 0.003 |
| N0-N1 | 59 | 58.31 (56.52–60.11) | |
| N2-N3 | 120 | 51.29 (48.16–54.42) | |
| **UICC/AJCC stage** | | | 0.086 |
| I-II | 20 | 59.60 (58.84–60.36) | |
| III-IVA/IVB | 159 | 52.87 (50.38–55.35) | |
| **PLR** | | | 0.001 |
| ≤130.7 | 105 | 56.76 (54.76–58.76) | |
| >130.7 | 74 | 49.21 (44.84–53.59) | |
| **LMR** | | | 0.001 |
| ≤3.3 | 62 | 48.80 (44.00–53.60) | |
| >3.3 | 117 | 56.19 (54.06–58.32) | |
| **HB** | | | 0.010 |
| ≤14.2 g/dL | 96 | 50.88 (47.26–54.49) | |
| >14.2 g/dL | 83 | 56.83 (54.65–59.01) | |
| **ALB** | | | <0.001 |
| ≤37.2 g/L | 23 | 41.40 (32.02–50.77) | |
| >37.2 g/L | 156 | 55.43 (53.43–57.43) | |
| **GLO** | | | <0.001 |
| ≤35.8 g/L | 145 | 55.26 (53.05–57.47) | |
| >35.8 g/L | 34 | 46.40 (39.88–52.92) | |

Notes.

CI, Confidence Interval; UICC/AJCC, Union for International Cancer Control/American Joint Committee on Cancer; PLR, Platelet-to-lymphocyte ratio; LMR, Lymphocyte-to-monocyte ratio; HB, Hemoglobin; ALB, Albumin; GLO, Globulin.

## A novel hematological parameter (GLO-PLR) was developed by combining independent risk factors

The multivariate analyses showed that PLR and GLO are independent predictors of prognosis in patients with NPC. Thus, we proposed a novel parameter GLO-PLR, which is a combination of GLO and PLR. GLO-PLR is calculated as follows: patients with GLO ≤35.8 g/L and PLR ≤130.7 are allocated to the low-risk group and assigned 0 points; patients with GLO ≤35.8 g/L and PLR of > 130.7 are allocated to the intermediate-risk group and assigned 1 point; patients with GLO> 35.8 g/L and PLR ≤130.7 are allocated to

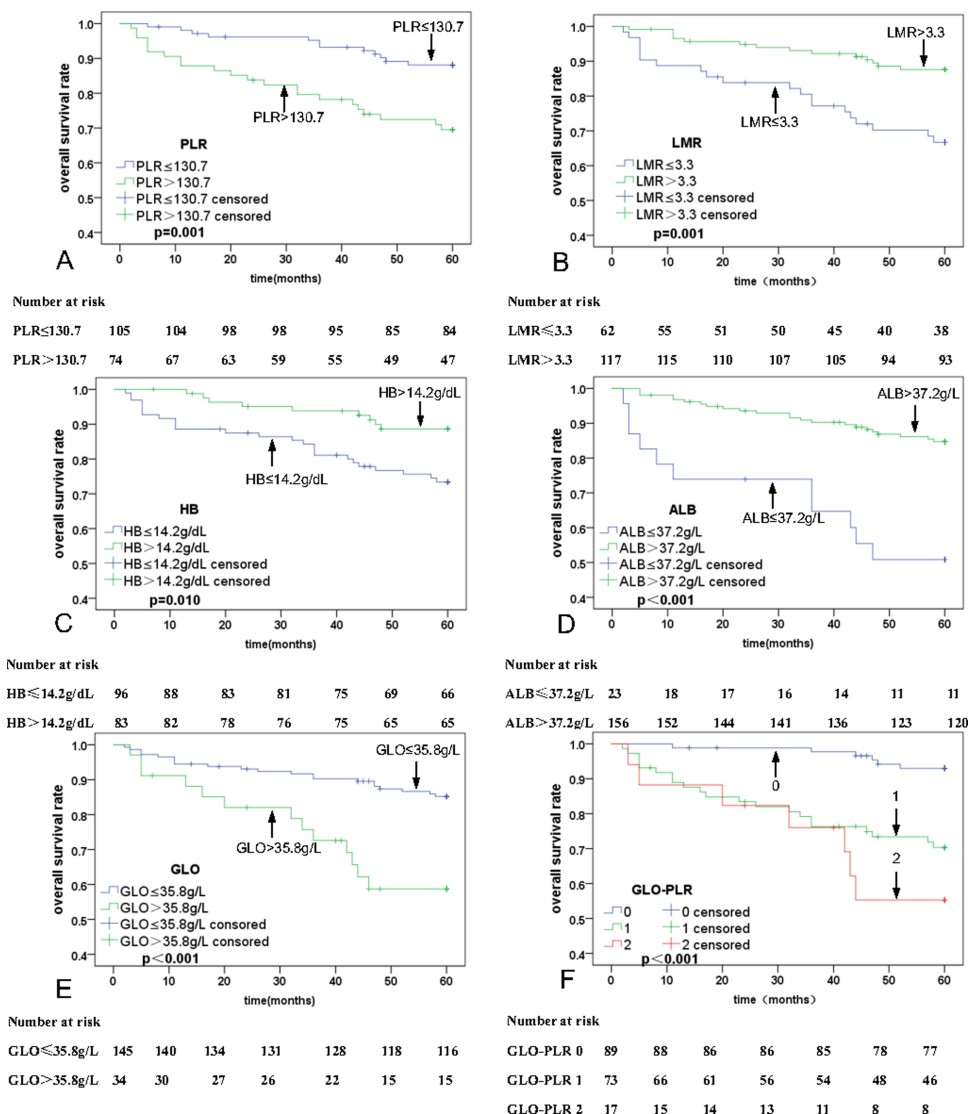

**Figure 2 Kaplan–Meier curves for overall survival (OS) according to PLR, LMR, HB, ALB, GLO, and GLO-PLR.** (A) OS stratified by PLR. (B) OS stratified by LMR.

the intermediate-risk group and assigned 1 point; patients with GLO > 35.8 g/L and PLR > 130.7 are allocated to the high-risk group and assigned 2 points. Among the 179 patients, 89 (49.7%) are assigned to the low-risk group, 73 (40.8%) are assigned to the medium-risk group, and 17 (9.5%) are assigned to the high-risk group.

The 5-year OS rates for patients with NPC in the low-risk, middle-risk, and high-risk groups are 93.9% ± 1.4%, 71.2% ± 10.8%, and 58.3% ± 25.1%, respectively. The 5-year PFS rates for patients with NPC in the low-risk, middle-risk, and high-risk groups are 89.9% ± 6.25%, 67.9% ± 10.9%, and 52.3% ± 25.3%, respectively. Thus, GLO-PLR is a significant prognostic factor of five-year OS and PFS in patients with NPC (Table 2, Figs. 2F, and 3F).

**Table 3** Univariate and multivariate analyses of clinicopathological characteristic for the prediction of overall survival in patients with nasopharyngeal carcinoma.

| Parameter | Univariate analysis | | multivariate analysis | |
|---|---|---|---|---|
| | HR (95% CI) | *P* | HR (95% CI) | *P* |
| **PLR** | | 0.001 | | 0.03 |
| ≤130.7 | 0.34 (0.17–0.68) | | 0.44 (0.21–0.91) | |
| >130.7 | 1 | | 1 | |
| **LMR** | | 0.001 | | 0.32 |
| ≤3.3 | 0.33 (0.16–0.64) | | 0.65 (0.28–1.52) | |
| >3.3 | 1 | | 1 | |
| **HB** | | 0.010 | | 0.44 |
| ≤14.2 g/dL | 2.61 (1.22–5.60) | | 1.41 (0.59–3.37) | |
| >14.2 g/dL | 1 | | 1 | |
| **ALB** | | <0.001 | | 0.02 |
| ≤37.2 g/L | 4.25 (2.07–8.74) | | 2.49 (1.17–5.30) | |
| >37.2 g/L | 1 | | 1 | |
| **GLO** | | <0.001 | | 0.02 |
| ≤35.8 g/L | 0.30 (0.15–0.60) | | 0.44 (0.21–0.90) | |
| >35.8 g/L | 1 | | 1 | |

Notes.

HR, hazard ratio; 95% CI, 95% Confidence Interval; UICC/AJCC, Union for International Cancer Control/American Joint Committee on Cancer; PLR, Platelet-to-lymphocyte ratio; LMR, Lymphocyte-to-monocyte ratio; HB, Hemoglobin; ALB, Albumin; GLO, Globulin.

## The relationship between GLO-PLR and clinical characteristics

There is no statistically significant difference in the GLO-PLR related to the distribution of sex ($p = 0.160$), age ($p = 0.068$), tissue type ($p = 0.724$), or treatment plan ($p = 0.767$). However, GLO-PLR is significantly associated with the T stage ($p = 0.002$), N stage ($p = 0.027$), and clinical stage ($p = 0.012$).

## Comparison of the AUCs for the GLO, PLR, and GLO-PLR

The discriminatory abilities of GLO, PLR, and GLO-PLR in predicting the prognosis are evaluated by using the ROC AUC. As shown in Fig. 4 and Table 6, the AUC of GLO-PLR for predicting OS is 0.714 (95% CI [0.621–0.806], $p < 0.001$), which is higher than those of GLO (AUC: 0.622, 95% CI [0.506–0.731], $p = 0.031$) and PLR (AUC: 0.644 95% CI [0.541–0.748], $p = 0.009$). As shown in Fig. 5 and Table 7, the AUC of GLO-PLR for predicting PFS is 0.696 (95% CI [0.605–0.786], $p < 0.001$), which is higher than those of GLO (AUC: 0.630, 95% CI [0.526–0.734], $p = 0.012$) and PLR (AUC: 0.627, 95% CI [0.539–0.735], $p = 0.013$).

## DISCUSSION

Concurrent radiochemotherapy is the major treatment option for patients with locally advanced NPC. Several studies have explored the potential benefits of adding neoadjuvant chemotherapy and/or adjuvant chemotherapy. However, studies have shown that not all patients require neoadjuvant chemotherapy and/or adjuvant chemotherapy. Therefore, we

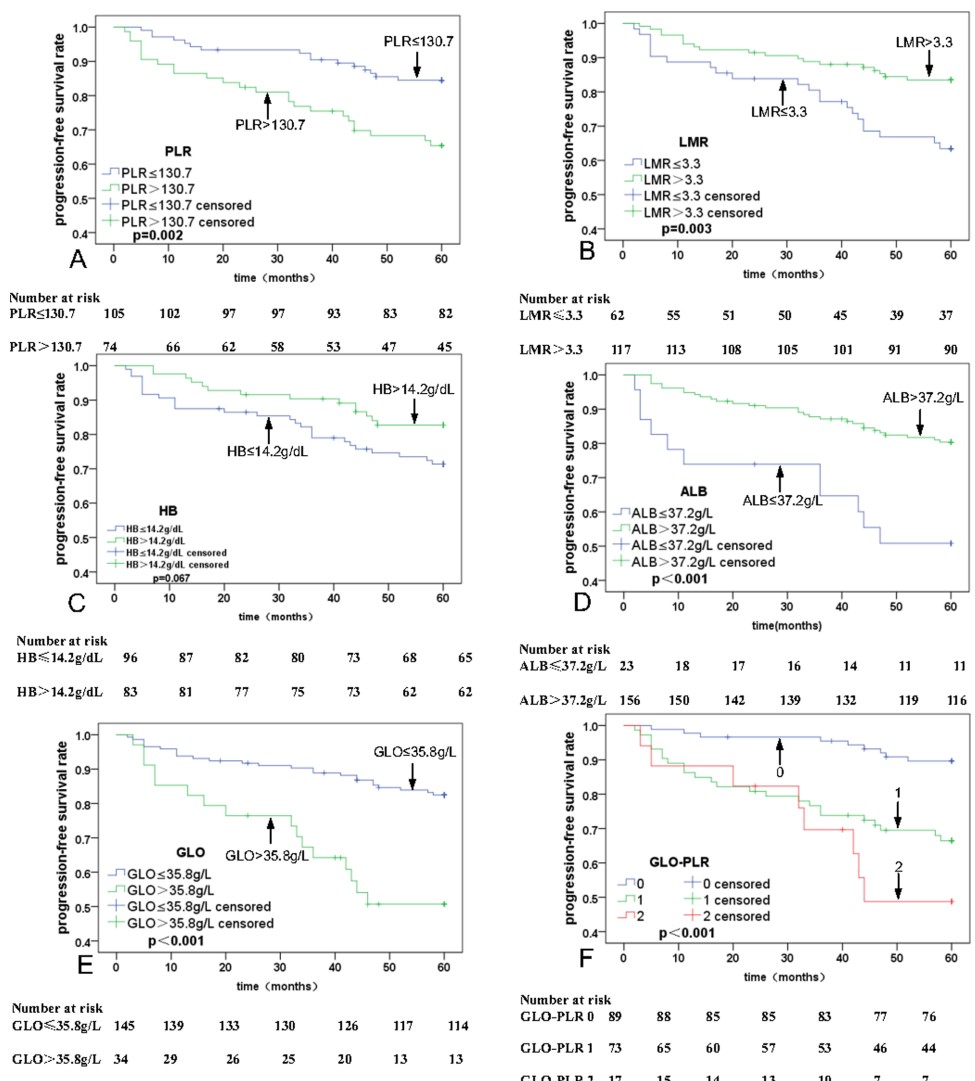

**Figure 3** Kaplan–Meier curves for progression-free survival (PFS) according to PLR, LMR, HB, ALB, GLO, and GLO-PLR. (A) PFS stratified by PLR. (B) PFS stratified by LMR.

need to identify biomarkers that can predict the prognosis of patients with NPC and help stratify high-risk groups to select the appropriate patients for neoadjuvant chemotherapy and/or adjuvant chemotherapy. This will facilitate the true individualization of therapy to offer maximal benefits. In this study, we explored values of hematological parameters before treatment as independent prognostic factors for patients' survival. In addition, this study proposed a new hematological parameter named GLO-PLR which stratifies patients according to the combination of two independent prognostic factors to better predict their outcomes.

Recent studies have shown that systemic inflammatory responses are associated with tumor progression and metastasis by inhibiting apoptosis, damaging DNA, and enhancing angiogenesis (*Hanahan & Weinberg, 2011*; *McMillan, 2009*). PLR is considered to be

**Table 4** Analysis of clinicopathological parameters for the prediction of progression-free survival in patients with nasopharyngeal carcinoma.

| Characteristics | No. of patients | S.T. (Month) Mean (95%, CI) | p-value |
|---|---|---|---|
| **Sex** | | | 0.631 |
| Male | 134 | 52.08 (49.29–54.88) | |
| Female | 45 | 52.35 (47.39–57.31) | |
| **Age at diagnosis, y** | | | 0.209 |
| ≤50 | 52 | 55.00 (51.39–58.60) | |
| >50 | 127 | 50.99 (47.92–54.59) | |
| **Tumor classification** | | | 0.164 |
| T0-T2 | 78 | 53.81(50.50–57.12) | |
| T3-T4 | 101 | 50.88(47.43–54.34) | |
| **Node classification** | | | 0.001 |
| N0-N1 | 59 | 58.00 (56.12–59.87) | |
| N2-N3 | 120 | 49.26 (45.87–52.66) | |
| **UICC/AJCC stage** | | | 0.045 |
| I-II | 20 | 59.60 (58.84–60.36) | |
| III-IVA/IVB | 159 | 52.21 (48.51–54.59) | |
| **PLR** | | | 0.003 |
| ≤130.7 | 105 | 56.17 (52.66–57.67) | |
| >130.7 | 74 | 47.87 (44.35–52.39) | |
| **LMR** | | | 0.004 |
| ≤3.3 | 62 | 48.20 (43.42–52.99) | |
| >3.3 | 117 | 54.23 (51.57–56.88) | |
| **HB** | | | 0.670 |
| ≤14.2 g/dL | 96 | 50.02 (46.28–53.75) | |
| >14.2 g/dL | 83 | 54.63 (51.73–57.521) | |
| **ALB** | | | <0.001 |
| ≤37.2 g/L | 23 | 41.40 (32.02–50.77) | |
| >37.2 g/L | 156 | 53.73 (51.41–56.06) | |
| **GLO** | | | <0.001 |
| ≤35.8 g/L | 145 | 54.33 (51.95–56.71) | |
| >35.8 g/L | 34 | 42.71 (35.69–49.73) | |

**Notes.**

CI, Confidence Interval; UICC/AJCC, Union for International Cancer Control/American Joint Committee on Cancer; PLR, Platelet-to-lymphocyte ratio; LMR, Lymphocyte-to-monocyte ratio; HB, Hemoglobin; ALB, Albumin; GLO, Globulin.

an accurate, easily detectable, and inexpensive marker for evaluating the prognosis of patients with NPC and guiding treatment decisions. In fact, elevated PLR levels can predict the prognosis of various tumors such as lung cancer (*Sanli et al., 2024*), esophageal cancer (*Aoyama et al., 2022*), and cervical cancer (*Gao et al., 2023*).

Although the specific mechanism of PLR in predicting prognosis has not yet been proven, several studies have found that platelets can protect tumor cells from degradation. The interaction between direct platelet-tumor cells and platelet-derived TGF$\beta$ can synergistically activate the NF-$\kappa$B and TGF$\beta$/Smad pathways in cancer cells, induce

**Table 5  Univariate and multivariate analyses of clinicopathological characteristic for the prediction of progression-free survival in patients with nasopharyngeal carcinoma.**

| Parameter | Univariate analysis | | multivariate analysis | |
|---|---|---|---|---|
| | HR (95% CI) | *P* | HR (95% CI) | *P* |
| **PLR** | | 0.003 | | 0.022 |
| ≤130.7 | 0.39 (0.21–0.73) | | 0.47 (0.25–0.90) | |
| >130.7 | 1 | | 1 | |
| **LMR** | | 0.004 | | 0.605 |
| ≤3.3 | 0.41 (0.22–0.75) | | 0.82 (0.38–1.77) | |
| >3.3 | 1 | | 1 | |
| **ALB** | | <0.001 | | 0.200 |
| ≤37.2 g/L | 3.19 (1.60–6.37) | | 1.71 (0.76–3.86) | |
| >37.2 g/L | 1 | | 1 | |
| **GLO** | | <0.001 | | 0.001 |
| ≤35.8 g/L | 0.28 (0.15–0.53) | | 0.37 (0.19–0.72) | |
| >35.8 g/L | 1 | | 1 | |

**Notes.**

HR, Hazard ratio; 95% CI, 95% Confidence Interval; UICC/AJCC, Union for International Cancer Control/American Joint Committee on Cancer; PLR, Platelet-to-lymphocyte ratio; LMR, Lymphocyte-to-monocyte ratio; HB, Hemoglobin; ALB, Albumin; GLO, Globulin.

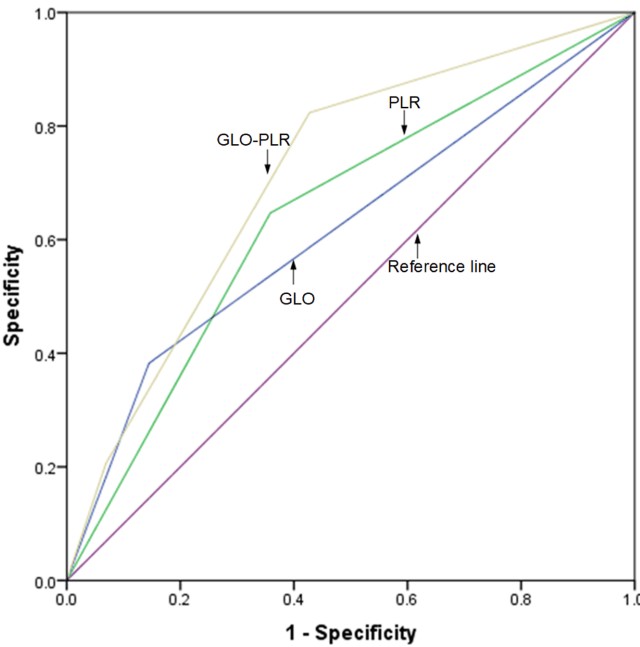

**Figure 4  Comparison of the discriminatory abilities of GLO, PLR, and GLO-PLR for predicting overall survival (OS).**

the transformation of epithelial cells to mesenchymal cells, and enhance metastasis *in vivo* (*Labelle, Begum & Hynes, 2011*; *Tesfamariam, 2016*). The secretion of VEGF and platelet-derived growth factor can promote angiogenesis and metastasis, thereby promoting

**Table 6   Comparison of the areas under the curves (AUCs) of overall survival for GLO, PLR and GLO-PLR.**

| Parameter | AUC | 95% CI | P |
|---|---|---|---|
| GLO | 0.619 | 0.506–0.731 | 0.031 |
| PLR | 0.644 | 0.541–0.748 | 0.009 |
| GLO-PLR | 0.714 | 0.621–0.806 | <0.001 |

Notes.

AUC, Area under curve; 95% CI, 95% Confidence Interval; GLO, Globulin; PLR, Platelet-to-lymphocyte ratio; GLO-PLR, Globulin and Platelet-to-lymphocyte ratio.

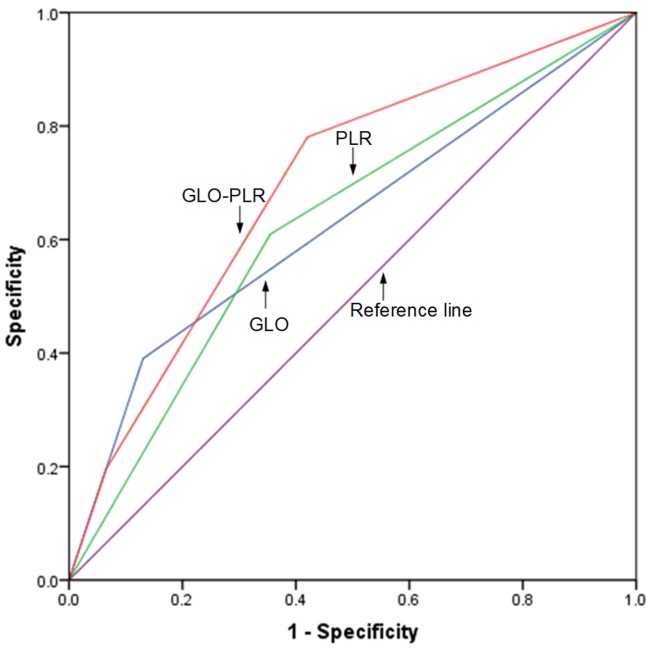

**Figure 5   Comparison of the discriminatory abilities of GLO, PLR, and GLO-PLR for predicting progression-free survival (PFS).**

**Table 7   Comparison of the areas under the curves (AUCs) of progression-free survival for GLO, PLR and GLO-PLR.**

| Parameter | AUC | 95% CI | P |
|---|---|---|---|
| GLO | 0.63 | 0.526–0.734 | 0.012 |
| PLR | 0.627 | 0.539–0.735 | 0.013 |
| GLO-PLR | 0.696 | 0.605–0.786 | <0.001 |

Notes.

AUC, Area under curve; 95% CI, 95% Confidence Interval; GLO, Globulin; PLR, Platelet-to-lymphocyte ratio; GLO-PLR, Globulin and Platelet-to-lymphocyte ratio.

the development and progression of tumors (*Bambace & Holmes, 2011*). In addition, tumor cell-induced platelet aggregation around the tumor can protect tumor cells from natural killer cells and TNF-$\alpha$-mediated cytotoxicity (*Philippe et al., 1993*; *Nieswandt et al., 1999*). Moreover, it can also increase tumour cell adhesion to endothelial cells and help tumor cell

extravasation (*Alonso-Escolano et al., 2006*). In addition, thrombopoietin can be produced after interleukin-6 (IL-6) stimulation, thereby increasing platelet counts in patients with tumors (*Lippitz, 2013*). Therefore, platelets are the key in the development and progression of tumors.

Lymphocytes can eliminate tumors through cell-mediated immune surveillance and anti-tumor immune responses. Moreover, T lymphocytes are related with a better prognosis in ovarian cancer (*Sato et al., 2005*), breast cancer (*Loi et al., 2013*), melanoma (*Clemente et al., 1996*), and colon cancer (*Galon et al., 2006*). Tumor-infiltrating lymphocytes (TILs) are differentially involved in various stages of tumor development, and they play an essential role in the composition of the tumor immune microenvironment. Studies have shown that refractory metastatic melanoma patients that receive cell infusion with autologous tumor-reactive, rapidly expanded TIL cultures, and high-dose Interleukin-2 (IL-2) therapy have improved outcomes. More than 50% of patients are resistant to this adoptive cell transfer immunotherapy (*Dudley et al., 2005*). In patients with NPC, almost all non-keratinizing undifferentiated nasopharyngeal carcinomas are associated with EBV infection (*Tao & Chan, 2007*). Moreover, EBV-specific cytotoxic T lymphocytes can mediate significant tumor regression in advanced NPC (*Straathof et al., 2005*; *Comoli et al., 2005*). With further research, lymphocytes have great potential in the clinical treatment and prognostic predictions of NPC.

PLR is the devision of platelet count by the lymphocyte count. Several researchers believe that PLR represents a balanced state between the tumor platelet-activating factor and inflammatory response factor lymphocytes. Elevated PLR levels represent increased platelet and/or decreased lymphocyte counts. Therefore, this inflammatory response orients the reaction toward tumor promotion. On the contrary, decreased PLR indicates decreased platelets and/or increased lymphocytes, and an inflammatory response that orients the reaction toward tumor suppression. Indeed, PLR is believed to be associated with the clinical survival of patients with NPC (*Cen & Li, 2019*). Similarly, our study found that the 5-year OS rate of patients with NPC in the low PLR group (PLR $\leq$ 130.7) is higher than that in the high PLR group (PLR> 130.7) (HR = 0.44, 95% CI [0.21–0.91], $p = 0.029$), and the 5-year PFS rate of patients in the low PLR group (PLR $\leq$ 130.7) is higher than that in the high PLR group (PLR> 130.7) (HR = 0.47, 95% CI [0.25–0.90], $p = 0.022$). Therefore, PLR may influence the tumor microenvironment and immune system status, thereby affecting the survival of patients with NPC.

Malnutrition is highly prevalent in cancer patients and often considered during cancer treatment. ALB is an important component of serum proteins and a representative marker of systemic inflammation (*Fernández et al., 2019*). Hypoproteinemia is mainly caused by reduced ALB synthesis and increased vascular permeability. However, an increase of ALB degradation is also a major cause of hypoproteinemia in cancer patients (*Fearon et al., 1998*). Therefore, ALB is often used to assess the nutritional status of patients. When patients suffer from malnutrition, it often leads to a decrease of humoral and cellular immunity, thereby increasing the risk of infection and reducing the effect of treatment (*Chandra, 1999*). ALB has also been used to predict the prognosis of multiple tumors (*Gupta & Lis, 2010b*). Previous studies have shown that reduced ALB concentration is associated with

poor prognosis in patients with NPC. The prognosis of the patients in the low ALB group (ALB $\leq$ 43 g/L) is worse than that in the high ALB group (ALB > 43 g/L) (*Li et al., 2014*). Our study shows similar results. The 5-year OS of the patients with NPC in the low ALB group (ALB $\leq$ 37.2 g/L) is worse than that in the high ALB group (ALB > 37.2 g/L) (HR =2.49, 95% CI [1.17–5.30], $p = 0.018$). However, the PFS analysis did not corroborate the above conclusions. The concentration of ALB is affected by various factors, such as hydration status, liver function, and kidney function of patients. Therefore, serum ALB concentrations before treatment have limitations in predicting the prognosis of NPC. When using serum ALB before treatment to predict the prognosis of patients with NPC, the hydration status, liver function, and kidney function should be identified and analyzed first.

GLO is another important component of the serum proteins and a representative indicator of systemic inflammation (*Feng, Huang & Chen, 2014*). Globulins mainly include immunoglobulins and acute-phase proteins. These proteins reflect the exposure state of inflammation and promote tumor cell growth and proliferation (*Li et al., 2014*). NPC is a typical inflammation-associated malignancy. In the high-risk region for NPC, there is a high correlation between NPC and EBV infection. EBV induces immunoglobulin production. Studies have shown that the concentrations of immunoglobulins such as those against Epstein-Barr viral capsid antigens and early antigen-IgA are negatively correlated with the short-term and long-term survival of NPC patients (*Solinas et al., 2010*). In our study, we explored the relationship between GLO concentration and the prognosis of OS and PFS in patients with NPC. The 5-year OS rate in the low GLO group is higher than that in the high GLO group (GLO > 35.8 g/L) (HR = 0.44, 95% CI [0.21–0.90], $p = 0.024$). The 5-year PFS rate in the low GLO group is higher than that in the high GLO group (GLO > 35.8 g/L) (HR = 0.37, 95% CI [0.19–0.72], $p = 0.001$).

Based on the results of the multivariate analysis, we proposed a new hematological parameter, GLO-PLR, for predicting the prognoses of patients with NPC. Our results show that the prognosis for patients in the low-risk group (score 0) is significantly better than that in the middle-risk group (score 1) and the high-risk group (score 2). When we analyzed the associations between GLO-PLR and basic clinical characteristics, we found that GLO-PLR is significantly correlated with T staging ($p = 0.002$), N staging ($p = 0.027$), and clinical staging ($p = 0.012$) in patients with NPC. This helps us to better understand the important role of GLO-PLR in NPC. We also compared the AUC values of the GLO-PLR parameter with those of the GLO and PLR to assess their ability to predict the prognosis of NPC. In predicting OS, the AUC of GLO-PLR is 0.714 (95% CI [0.621–0.806], $p < 0.001$), which is higher than the AUC of GLO and PLR. In predicting PFS, the AUC of GLO-PLR is 0.696 (95% CI [0.605–0.786], $p < 0.001$), which is higher than the AUCs of GLO and PLR. Therefore, we found that GLO-PLR is a new and accurate hematological parameter for predicting the prognosis of patients with NPC who received IMRT. In addition, GLO-PLR is an inexpensive, objective, and easy-to-measure marker.

This study has certain limitations. First, the sample size of the study is relatively small, since we analyzed 179 patients. Second, the study only involves data from one research center. More data from other hospitals well be better for the results validation. Third,

the study only focuses on conventional hematological parameters. We do not compare GLO-PLR with traditional prognostic indicators, such as the DNA concentration of Epstein-Barr virus and lactate dehydrogenase. A large-scale prospective study may be meaningful for future work.

### Funding

The authors received no funding for this work.

### Competing Interests

The authors declare there are no competing interests.

### Author Contributions

- Wenhua Tang performed the experiments, analyzed the data, prepared figures and/or tables, authored or reviewed drafts of the article, and approved the final draft.
- Guoxian Long conceived and designed the experiments, prepared figures and/or tables, authored or reviewed drafts of the article, and approved the final draft.

### Human Ethics

The following information was supplied relating to ethical approvals (i.e., approving body and any reference numbers):

Tongji Hospital of Huazhong University of Science and Technology.

### Data Availability

The raw measurements are available in the Supplementary Files.

### Supplemental Information

Supplemental information for this article can be found online at http://dx.doi.org/10.7717/peerj.17573#supplemental-information.

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
