# Peer review of "Retrospective study of a novel hematological parameter for predicting the survival of patients with nasopharyngeal carcinoma"

_PeerJ, doi:10.7717/peerj.17573_

## Round 0.1 · original submission · Major Revisions

Reviewers provided valuable insights and offered impactful feedback on this manuscript. Kindly incorporate their suggestions and carefully address all comments in your manuscript. Thank you

Reviewer 1 ·

Basic reporting

This is an interesting study where the researchers studied the prognostic values of routine pre-treatment hematological parameters in patients with nasopharyngeal carcinoma. Some comments are as follows.
Abstract
Line 40 – 41. The two sentences “Pre-treatment PLR, ALB, and GLO are independent predictors of 5-year OS in patients with NPC. Pre-treatment PLR and GLO are independent predictors of 5-year FPS in patients with NPC” can be merged and made into one sentence.


Introduction:
Line 47 - 49. The reference is missing. Please add the references.
Line 52 – 52. Add references regarding radiotherapy and its indications, especially IMRT.
Line 54 – 57. Add references to the survival rates.
Add the latest reference on the TNM staging.
In the introduction, it is better to add regarding the chemotherapy. Chemoradiotherapy and/or adjuvant chemotherapy or neoadjuvant chemotherapy with references.
Add also the potential biochemical changes and side effects.

Discussion
Discuss the prevalence of NPC in various ages and genders.
One limitation of this research is that this retrospective study is very old. The authors need to provide a rationale or modify this research.
It is better to add a conclusion and future directions for this study.

Figures
The authors need to add the full form of PLR, LMR, HB, ALB, GLO, and GLO-PLR in the Figures.

Experimental design

Line 105. The ethical approval date and number are needed.
In the inclusion criteria, what about age and gender?
Line 119 – 120. Add references of the UICC/AJCC.
However, this research needs a major revision in the method as it is very old.

Validity of the findings

Line 21 - 22. The authors mentioned that they have collected the data from 2012 to 2013. The data are over 10 years old. The authors need to give a rationale behind the old data. Is it possible for the researchers to conduct using the recently collected data?

Additional comments

Overall:
The English is good.
The references are old and there are very less references from the last 5 years references. The authors need to add the latest 5-year references.

·

Basic reporting

The authors have made a good attempt at providing an elaborate introduction to the topic. However certain additions/modifications would go a long way in improving the understanding of the reader specially in terms of understanding the objectives of the study.


• The introduction section can include a description of the utility of the X-tile software in the determination of cut-off values for overall survival and progression free survival.
• Line 96 (last paragraph of the introduction) seems to be a conclusive statement of the previous paragraph (line 95). It is advised to merge these sentences within the previous paragraph itself. Also, an alternative term to “retrospective’ can be used in this sentence.
• The conclusive paragraph can highlight the parameters that were evaluated in this study as inexpensive alternatives and then go on to describe the objectives of the study. Also, highlight the introduction of the novel in this section.
• Kindly provide a rational for choosing GLO-PLR as a prognostic predictor in NPC patients.

Experimental design

The authors have made their best effort to explain in detail the various aspects undertaken to conduct this study. However, description of these aspects into various sub-sections would help the reader in understanding the finer aspects of the methodology allowing him/her to replicate this study easily.

• The methodology section is overall a little confusing to the reader. It is advised to structure the methodology section into the following sections:
o Study Design: Describe the type of the study and the parameters to be explored. The study site and duration of participant recruitment has already been mentioned which can be explored in this section. Also mention the follow up period of these participants. Include the statement of ethical approval in this section.
o Inclusion & Exclusion Criteria: Mention the sample number followed by the inclusion and exclusion criteria. The treatment details can be described in this section. Categorization of participants based on age, clinical staging can be mentioned in this section.
o Parameters evaluated: Give a description of all the parameters that were evaluated in this study. Also give description of how the novel parameter was calculated (at least the formula for determination of groups as described in the results section). Also include OS and PFS in this section only as these parameters are also studied.
o Determination of Cut-off values: describe the utilization of the X-tile software here and how the cut-off value is determined.
o Statistical Analysis: the statistical tests employed in this study will be described here.

Validity of the findings

The authors have done a good job in providing the study data in graphical and tabular forms. A brief summary of the key findings in written form (as a paragraph) would be welcome.
• Line 157-158: the total number of patients from stage I to stage IVA/B is not totaling up to 179. Kindly correlate with values given in table 1. Results section can highlight all the key findings of the study in a paragraph form which will be further displayed in the form of tables and figures.

Additional comments

• AUC has been repeated twice under the abbreviations section in page 6.
• Adherence to STROBE statement not met entirely.
o The title doesn’t reflect the study design. This seems to be a retrospective observational study.
o Elaborate each of the specific objectives in the last paragraph of the introduction.
o Study design can be better described as mentioned above. This would help address adhering to most of the aspects of the STROBE statement.

---

## Round 0.2 · accepted · Accept

The authors have conducted excellent work on this manuscript, incorporating feedback from reviewers to enhance the clarity and depth of the content. They have successfully elucidated the role of a Novel Hematological Parameter in Nasopharyngeal Carcinoma. Thanks to the thoughtful integration of the reviewers' suggestions.

Reviewer 1 ·

Basic reporting

Overall, the manuscript is improved and made clear.

Experimental design

Some concern in the experimental design is added and improved.

Validity of the findings

The findings are made clear.

Additional comments

The manuscript is acceptable to publish.